# Deep Learning-Based Classification of Raw Hydroacoustic Signal: A Review

**Xu Lin, Ruichun Dong and Zhichao Lv \***

College of Ocean Science and Engineering, Shandong University of Science and Technology, Qingdao 266590, China
\* Correspondence: lvzhichao@hrbeu.edu.cn

**Abstract:** Underwater target recognition is a research component that is crucial to realizing crewless underwater detection missions and has significant prospects in both civil and military applications. This paper provides a comprehensive description of the current stage of deep-learning methods with respect to raw hydroacoustic data classification, focusing mainly on the variety and recognition of vessels and environmental noise from raw hydroacoustic data. This work not only aims to describe the latest research progress in this field but also summarizes three main elements of the current stage of development: feature extraction in the time and frequency domains, data enhancement by neural networks, and feature classification based on deep learning. In this paper, we analyze and discuss the process of hydroacoustic signal processing; demonstrate that the method of feature fusion can be used in the pre-processing stage in classification and recognition algorithms based on raw hydroacoustic data, which can significantly improve target recognition accuracy; show that data enhancement algorithms can be used to improve the efficiency of recognition in complex environments in terms of deep learning network structure; and further discuss the field's future development directions.

**Keywords:** deep learning; spectrum analysis; hydroacoustic target recognition; data augmentation

## 1. Introduction

Earth observation, changes in the marine environment, and climate change have been the focus of human attention in recent years, and these areas significantly impact productive human life [1]. With the growing need for underwater detection, underwater target recognition has recently become an active research area, and it is widely used in the fields of the surveying and modeling of the aquatic environment [2], underwater target localization and identification [3], and ship noise classification [4]. The rapid development of artificial intelligence technologies such as machine learning and deep learning, the emergence of supercomputing, the significant increase in arithmetic power, and the explosive growth of big-data-processing algorithms bring new opportunities for underwater target recognition. The U.S. has rapidly researched new computing and sensor technologies such as artificial intelligence, deep learning, machine learning, predictive analytics, high-powered active sonar transmitters, and new towed-line array sonar telemetry components. The related researchers are rapidly incorporating the results of research to enhance technology iterations. However, the application of deep learning in underwater target recognition still faces problems, such as small data volumes, the poor adaptability of traditional visual network algorithms, and complicated pre-processing process, and the patterns of deep learning are still too intricate to provide excellent generalizability. We note that acoustic and various signal-filtering methods are effectively used in detecting pipeline leaks, and various deep-learning algorithms are also widely used, indicating that deep-learning models have good generalizability and promise in underwater acoustics [5].

In order to better explore the applicatory potential of deep learning in underwater target recognition, this paper provides a comprehensive description of underwater signal

recognition via deep learning based on the advantages of various types of neural networks and discusses the field's future development direction. Please refer to Figure 1 for various methods.

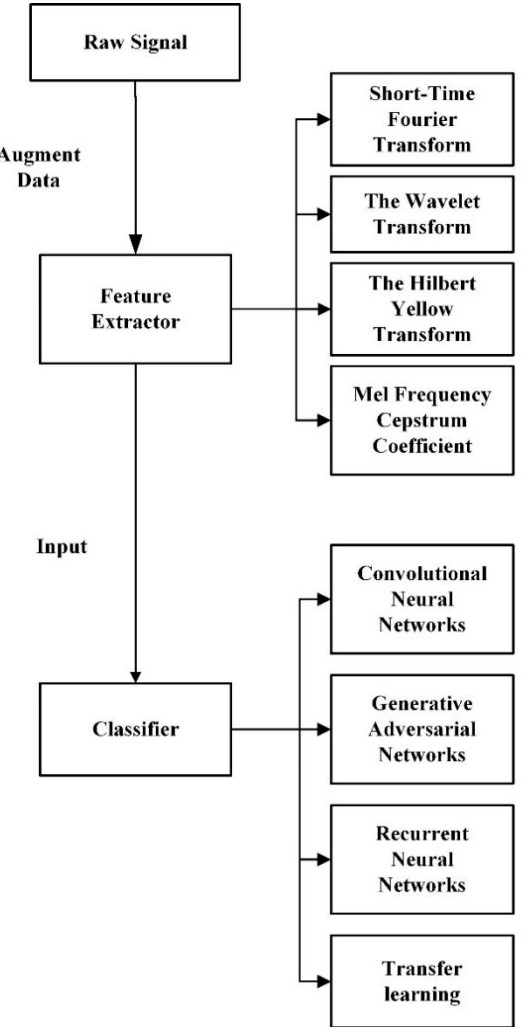

**Figure 1.** Classification process and methods.

The following are the main contributions of the article.

1. This paper clarifies and reviews various methods used for signal processing. This paper provides an overview of signal processing according to its developmental history, i.e., in terms of Fourier transform, short-time Fourier transform (STFT), Hilbert–Yellow transform, the Meier spectrum (MFCC), and wavelet transform (WT), and explores the advantages and disadvantages of various signal-processing methods and future development directions by comparing the structures of practical applications.
2. This paper introduces various neural networks used in hydroacoustic signal recognition. In addition, unsupervised adaptive methods based on sound signals, such as migration learning and adversarial learning, are proposed in this paper.
3. This paper also provides an in-depth analysis of the problems of hydroacoustic signal processing and the corresponding direction of future development by comparing this method with data enhancement methods.

## 2. Raw Signal

The raw signals received by hydrophones are generally saved in a WAV format, as exemplified by the ShipsEar dataset [6], which was released in 2016 and contained 90 hydroacoustic audio tracks representing 11 vessel sounds as marine environmental noise. This dataset is saved in a WAV format to develop and test hydroacoustic applications. Therefore, it is one of the keys to retaining the full features and an optimal degree of denoising when extracting features from a signal for hydroacoustic signal recognition. This dataset is saved in a WAV format for developing and testing hydroacoustic applications.

### 2.1. The Fourier Transform

Fourier transform [6] represents the ability to convey a function that satisfies certain conditions as a linear combination of trigonometric functions (sine and cosine functions) or their integrals. In audio processing, Fourier transform is used as a method for analyzing a signal by examining its components and using them to synthesize the signal. In the analysis of signals, its main application is in processing smooth signals. The Fourier transform method allows researchers to obtain the general frequency components that a segment of the signal contains.

An accurate and efficient method for parameter estimation is proposed in [7] using the fractional order Fourier transform (FFT) method. The algorithm proceeds iteratively, returning the parameter estimates of the most dominant signal components, and effectively reducing the computational effort.

However, the moment of appearance of each component is not known. Thus, for non-smooth signals, the Fourier transform method shows its limitations. Therefore, various new spectral processing methods have been proposed to address the boundaries of the Fourier transform method.

### 2.2. The LOFAR Spectrum

The LOFAR spectrum is a power spectrum obtained by the original signal's short-time Fourier transform (STFT). Due to the non-stationary nature of the noise signal, its signal characteristics will also change significantly with time, so the Fourier transform method cannot be perfectly adapted to such problems. In order to overcome the issue of the non-smoothness of the signal, the LOFAR spectrum is widely used in hydroacoustic signal identification, which can reflect the characteristics of two dimensions of the time domain and frequency domain.

Chen et al. [8], based on LOFAR spectrum enhancement for underwater target recognition, designed a multi-step decision-algorithm-based enhancement method with which to recover the breakpoints in the LOFAR spectrum, which showed an excellent recognition rate in the CNN network. Jin et al. [9] compared standard spectrum maps, such as LOFAR, Audio, Demon, Histogram, etc.; implemented these spectrograms into the AlexNet network; and found that the LOFAR spectra had the highest recognition rate. From Figure 2, it can be found that the LOFAR spectrum has more prominent features than the original audio.

AlexNet [10] is a CNN (Convolutional Neural Networks) network structure designed by Hinton, the winner of the 2012 ImagNet competition, and his student Alex Krizhevsky, which will be described in detail below. We believe that although the LOFAR spectrum has some limitations, subsequent improvements in other spectra can adequately compensate for the shortcomings of the LOFAR spectrum, which leads us to the conjecture of multidimensional feature fusion. The concept of multidimensional feature fusion will be mentioned several times.

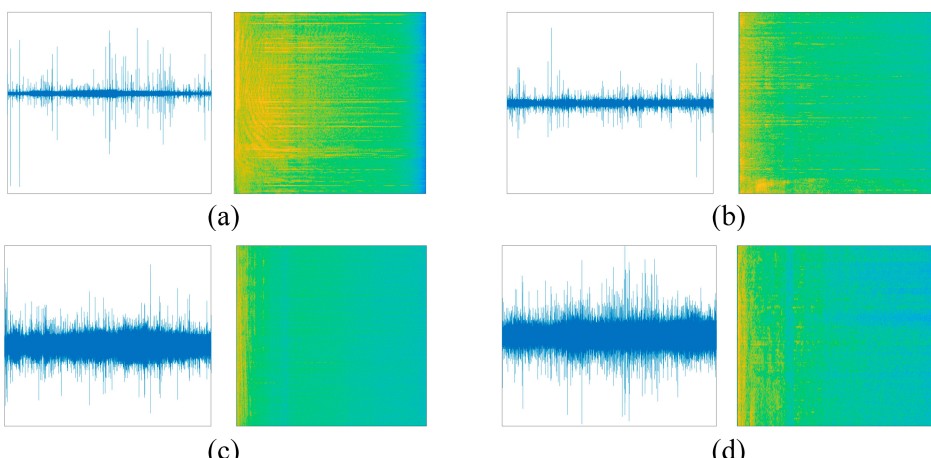

**Figure 2.** Comparison of the four LOFAR spectra with the original audio images. ((**a**) Excerpt from underwater audio of Yate, (**b**) excerpt from underwater audio of pesqSaladinoPrimero, (**c**) excerpt from underwater audio of corriente, and (**d**) excerpt from underwater audio of Draga.)

### *2.3. The Wavelet Transform*

The wavelet transform (WT) [11] is a development of the short-time Fourier transform, which inherits and develops the idea of the localization of the short-time Fourier transform while overcoming the drawback in which the window size does not change with frequency and can provide a "time–frequency" window that changes with frequency, with a lower time resolution and higher frequency resolution in the low-frequency part, and a higher time resolution and lower frequency resolution in the high-frequency part, which is very suitable for analyzing non-stationary signals and extracting the local features of signals; thus, the wavelet transform is considered to be a microscope for analyzing and processing signals.

As early as 1998, Chen et al. [12] introduced the wavelet transform as a feature extractor for underwater signals. The wavelet transform yielded a significant improvement in feature recognition compared to the original signal's average power spectral density (APSD). In 2013, Li et al. [13] proposed an improved SPIHT algorithm for shortcomings in hydroacoustic images. Their comparison results showed that the improved adaptive algorithm offered significant advantages. In 2017, R. Priyadharsini et al. [14] proposed a wavelet transform-based contrast enhancement method for hydroacoustic images, which uses a smooth wavelet transform (SWT) to decompose the input image into four components, namely, Low–Low, Low–High, High–Low, and High–High components, to obtain a better compression ratio. The High–Low and High–High components along with the high-contrast image are reconstructed by applying a smooth inverse transform combining the enhanced LL components and other subbands. Moreover, in 2021, Qiao et al. [15] similarly used the local wavelet acoustic model to classify underwater targets, see Figure 3. Thus, we can see the wide range of wavelet transform-processing methods used in hydroacoustic signal recognition.

However, the wavelet transform method has its drawbacks. In [16], Michael Weeks and Magdy Bayoumi identified the drawbacks of various discrete wavelet transform (DWT) system architectures; in general, the excellent properties of the wavelet transform in one dimension cannot be extended to two dimensions or higher, and it exhibits a lack of adaptivity to other modal decomposition methods, e.g., EMD, LMD, VMD, SGMD, etc. Due to the complexity of the hydroacoustic environment, the one-dimensional feature vector extracted by the wavelet transform is often insufficient for providing optimal features; thus, the discovery of method with which to improve the selection of optimal features or multidimensional features has become a possible direction of development.

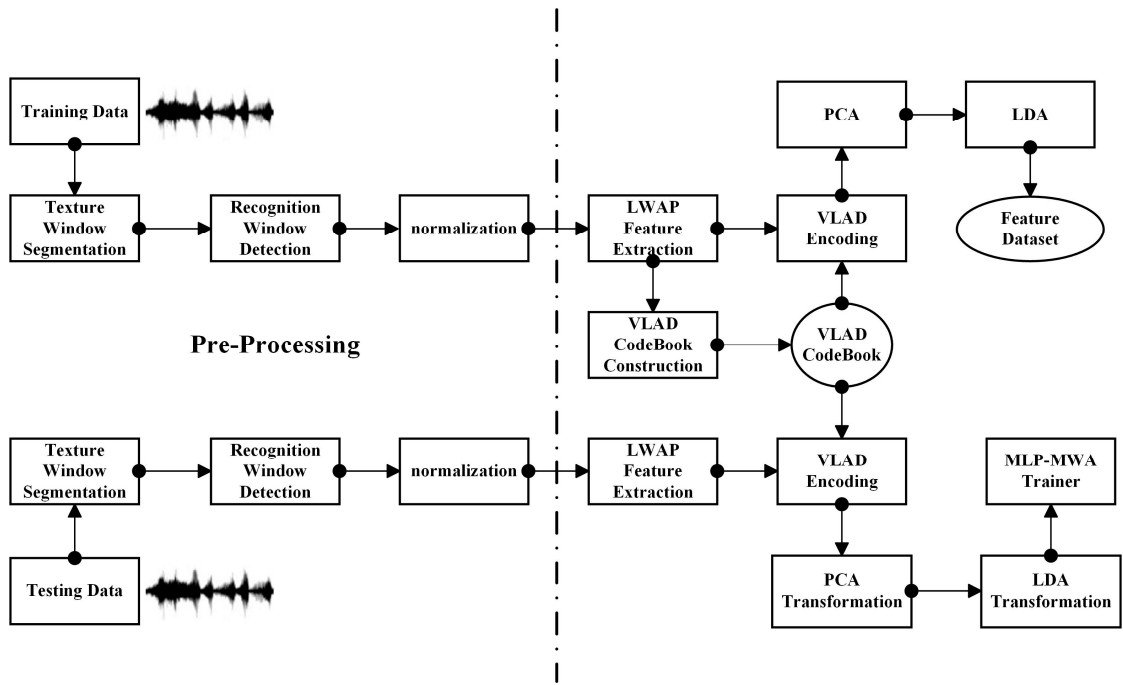

**Figure 3.** Passive sonar classifier flow proposed by Qiao et al.

*2.4. The Hilbert–Yellow Transform*

To cope with the shortcomings of wavelet transform methods in signal analysis, Norden E. Huang of NASA proposed a new data-processing method: the Hilbert–Huang transform (HHT) [17]. This transform consists of two parts: the empirical-mode decomposition (EMD) method [18] and the Hilbert transform. The main idea is to first decompose the original signal to a sum of finite eigenmode functions (IMFs) by using the empirical method for mode decompositions and then construct the corresponding analytical signal utilizing the Hilbert transformation, from which the instantaneous amplitude and instantaneous frequency of the time series are obtained in a second step. Below, we derive the Hilbert spectrum of the signal.

For a time function $x(t)$, defined with the interval $(-\infty < t < +\infty)$, the Hilbert transform of $x(t)$ and its inverse transform can be defined as follows:

$$y(t) = H[x(t)] = \frac{1}{\pi} P \int_{-\infty}^{\infty} \frac{x(\tau)}{t - \tau} d\tau = \frac{1}{\pi t} * x(\tau) \tag{1}$$

$$x(t) = H^{-1}[y(t)] = -\frac{1}{\pi} P \int_{-\infty}^{\infty} \frac{y(\tau)}{t - \tau} d\tau = -\frac{1}{\pi t} * y(\tau) \tag{2}$$

where $P$ denotes the principal Corsi value, $t$ and $\tau$ denote time, and the symbols $y(t)$ and $x(t)$ in the above equation form a pair of Hilbert transform pairs.

Unlike wavelet analysis, the EMD decomposition method performs decomposition via the signal's time-scale properties; in contrast, wavelet decomposition requires an a priori harmonic function and a wavelet function, so the Hilbert–Yellow transform is considerably superior in terms of objectivity and resolution.

In hydroacoustic signal recognition, the Hilbert–Yellow transform is often used to extract underwater target features [19]. In 2009, Liu et al. [20] proposed an improved HHT based on the limitations of the Hilbert–Yellow transform, and in this improved version, the instantaneous harmonic inversion method is used to calculate the instantaneous frequency of the eigenmodal function. Their simulation results show that the improved HHT has greater frequency resolution and more advantages thereof than the original HHT. In addition, experimental data proved that the improved version is effective for

hydroacoustic signal detection. In 2013, Wang et al. [21] proposed a time–frequency analysis method combining Barker wavelet analysis and Hilbert–Yellow transform to address the effect of environmental noise on recognition accuracy during long-distance detection. First, Barker wavelet analysis was used to divide the signal into subbands corresponding to auditory perception. Then, denoising was applied to enhance the analyzed signal; finally, the Hilbert–Yellow transform was used to extract the transient frequencies and amplitudes. Based on these transient parameters, various features were constructed and compared.

### 2.5. The Mel Spectrum

Before introducing the Meier spectrum, we should understand the concept of the Meier scale. The Mel scale [22] was named by Stanley Smith Stevens, John Volkmann, and Newman in 1937. Inspired by the reception of high- and low-frequency signals detectable by the human ear, the part of the Mel scale in the low-frequency band has an almost linear relationship to the average frequency, while in the high-frequency band, the two show a logarithmic relationship, which accurately simulates the characteristic wherein the human ear is more sensitive to the distinction between low-frequency signals and less sensitive to the distinction between high-frequency signals [23]. Finally, the Mel filter bank is multiplied by the power spectrum to obtain the Mel spectrum, see Figure 4.

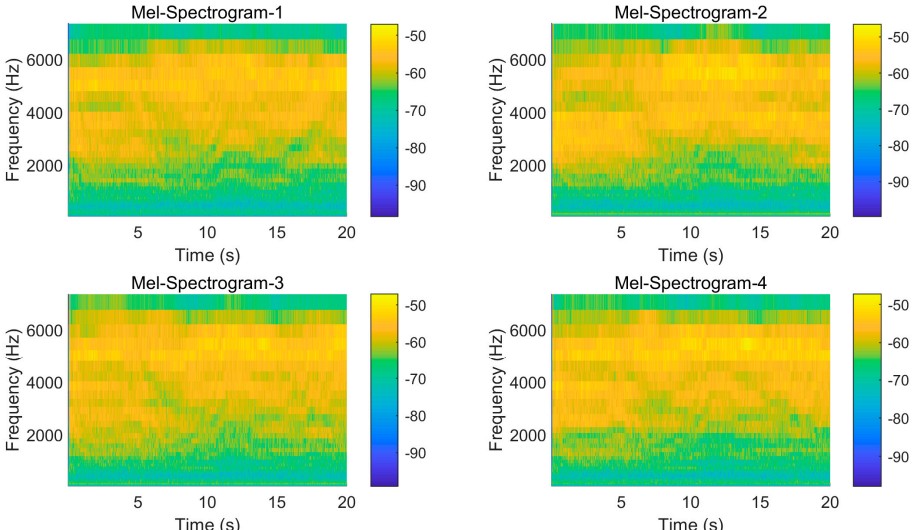

**Figure 4.** Mel Spectrum.( Sampling frequency 44.1kHz, the darker the color, the higher the absolute value of the decibel number (with negative sign)).

As mentioned above, the Mel spectrum is more sensitive to low-frequency signals, and the signal characteristics of marine organisms [13] and ship noise have the same significant features at the low-frequency end; thus, the Mel spectrum was introduced and widely used for hydroacoustic signal recognition. Liu et al. [14] constructed 3-D features using the Mel spectrum, delta and delta–delta features, and an input convolutional recurrent Neural Network (CRNN) for acoustic target recognition (Figure 5) and achieved a 94.6% correct recognition rate.

However, the impact of the Mel spectrum is twofold: on the one hand, it preserves the characteristic frequencies close to the human ear, while on the other hand, it loses resolution information; the fundamental reason for this is that the Mel coefficients are the energy sum of a filter band.

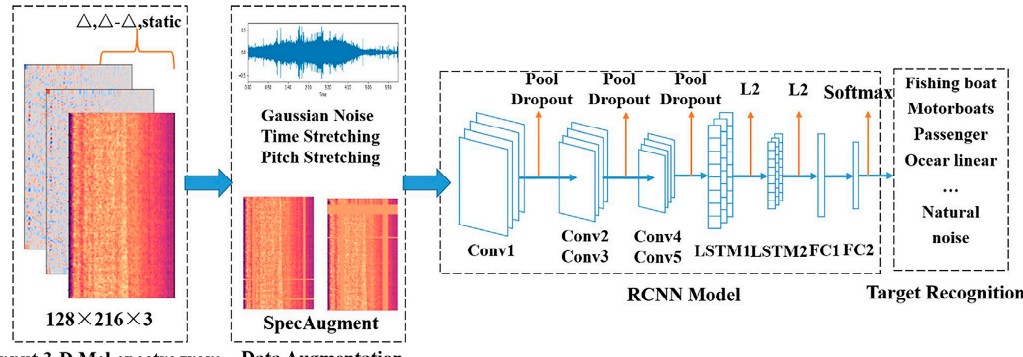

**Figure 5.** 3-D Meier spectrum-based classification proposed by Liu et al.

We can also obtain another inverse coefficient, the Mel Frequency Cepstrum Coefficient (MFCC), based on the Meier spectrum. The MFCC is a spectrum that can be used to represent short-term tones and is based on the principle of a logarithmic spectrum expressed in a nonlinear Meier scale and its linear cosine transformation. The MFCC is also widely used to extract audio features.

It was verified in the literature [24] that MFCC, first-order differential MFCC, and second-order differential MFCC features can all be used as practical features for identifying underwater targets. The feasibility of Mel frequency cepstrum coefficients in ship-noise feature extraction was similarly verified in [25]. Based on the processing of the Meier spectrum, the discovery of a method with which to retain more original features has become one of the developmental directions of hydroacoustic signal recognition.

### 2.6. Feature Fusion

There are many methods for acoustic feature fusion, of which all involve essentially two identical features flowing into the network simultaneously, splicing the same frame forward and updating both branches simultaneously during backward propagation. The standard acoustic feature fusion methods are as follows: (1) splicing based on the acoustic feature itself, such as MFCC + pitch [26]; (2) deep feature fusion in offline conditions, for example, an early bottleneck feature and MFCC splicing as a tandem feature, first using the network to extract the bottleneck feature and then splicing with MFCC to form new features for later model training [27]; and (3) feature fusion in online conditions [28].

Zhang et al. [29] proposed an integrated neural network using a short-time Fourier transform magnitude spectrum and short-time Fourier phase spectrum feature fusion learning. Their study showed that the integrated neural network method based on feature fusion has a higher recognition rate and noise robustness, see Figure 6. Whereas B. Mishachandar et al. [30] utilized the Meier spectrum, Meier cepstrum, and the LOFAR spectrum as multi-dimensional vector feature inputs; their proposed method can self-learn features from the data, eliminating the feature extraction step.

In general, due to each type of spectrum's shortcomings, the processing of a single spectrum cannot meet the neural network's requirements for the extracted features. The feature fusion approach compensates for the lack of hydroacoustic data sets on the one hand and improves the network efficiency on the other; we believe that the study of more feature fusion methods will be one of the developmental directions of hydroacoustic information processing.

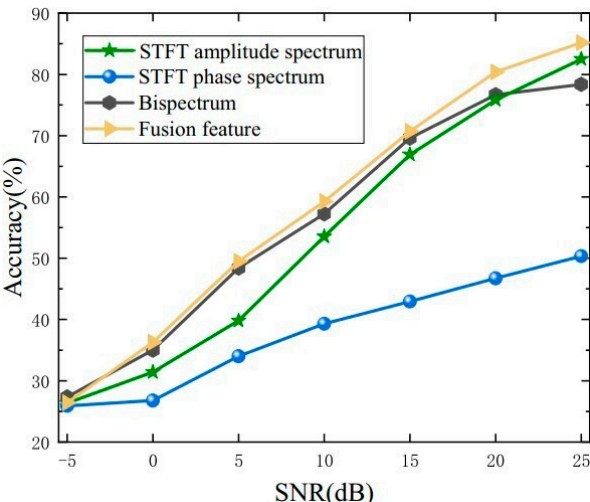

**Figure 6.** Feature recognition results of the method proposed by Zhang et al. [29] at different signal-to-noise ratios.

### 3. Deep Learning-Based Hydroacoustic Signal Recognition

Since the non-uniformity of a seawater medium can cause the attenuation and distortion of acoustic signals, while various floating objects and particles can increase the multipath effect during acoustic wave transmission, it is often difficult to achieve good recognition-related results using the traditional method for hydroacoustic signal recognition [31–33]. At the same time, deep learning is based on artificial neural networks and consists of multiple processing layers with which to study data with different levels of abstraction. It can be adequately applied to structured and unstructured data, and deep learning is widely used in new fields such as hydroacoustic signal recognition [34,35].

Deep-learning algorithms can be classified into various forms, such as supervised, semi-supervised, and unsupervised. Supervised learning methods are based on the training of models using correctly classified data or labels; supervised algorithms analyze the training data and produce an inferred function that can be used to map new examples. An optimal solution would allow the algorithm to correctly determine class labels when the labels are not visible [36]. Unsupervised learning functions are used for unsupervised datasets and to solve various problems in pattern recognition based on training samples of unknown classes in cases where manual labeling is complex [37]. Semi-supervised learning is a learning method that combines supervised learning with unsupervised learning. Semi-supervised learning simultaneously uses vast quantities of unlabeled data and labeled data to perform pattern recognition [38]. Another deep-learning algorithm is migration learning, which is used to improve a model from another domain by migrating information from a related domain [39], wherein hydroacoustic data are very scarce, while migration learning effectively solves the problem of insufficient hydroacoustic data.

Deep learning can also solve the problem of a loss of hydroacoustic detection echo features. Li et al. [40] proposed a Bidirectional Long- and Short-Term Memory Neural Network (Bi-LSTM) method based on vector sensors and without using pre-extracted features and applied it to the field of hydroacoustics for the first time; then, they compared Bi-LSTM and the difference between LSTM and the support vector machine and the influence of some parameters on the recognition rate. Afterwards, the method's robustness was verified by navigation tests conducted to achieve effective target recognition.

The advantages of deep learning discussed above have led to its widespread application in areas such as bracketing images, speech and text recognition, target detection, pattern recognition, and fault and medical anomaly diagnosis. In hydroacoustic signal recognition, deep learning shows its superiority over traditional approaches; next, we will introduce standard deep-learning models and their applications in hydroacoustic signal recognition.

### 3.1. Convolutional Neural Networks (CNN)

Research on convolutional neural networks began in the 1980s and 1990s. Time-delay networks and LeNet-5 were the first convolutional neural networks to appear [41]. LE-CUNN was first proposed for image processing in 1998 [42]; since then, the proposed deep-learning theory and the improvement of numerical computing devices and convolutional neural networks have developed rapidly. Convolutional neural networks are widely used in computer vision, natural language processing, target recognition, medical prediction, and other fields.

Since the introduction of the AlexNet network, Figure 7, which was discussed earlier, convolutional neural networks have entered a period of rapid development, and deep convolutional neural networks have rapidly replaced traditional image classification and recognition methods over a short period. CNNs consist of a multilayered structure, including convolutional, nonlinear, pooling, and fully connected layers. Due to their convolutional and pooling operations and parameter sharing, which enable deep learning structures to operate in a variety of devices, CNNs excel in machine learning problems, especially for applications dealing with image data [43,44].

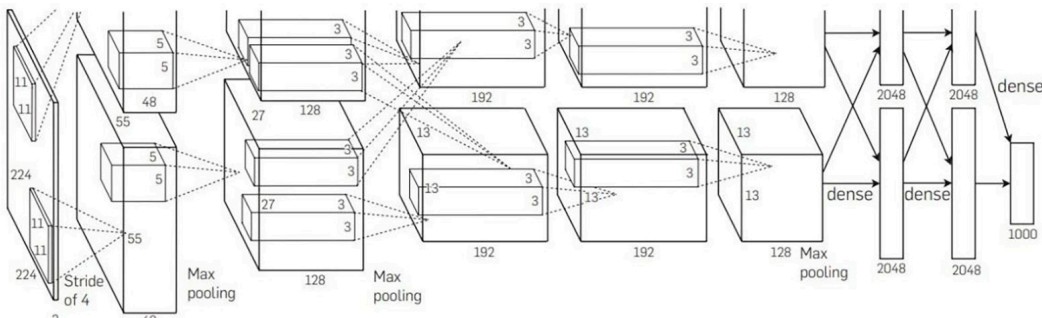

**Figure 7.** Structure of AlexNet network [10]. (Reprinted/adapted with permission from Ref. [ImageNet classification with deep convolutional neural networks]. A. Krizhevsky, Ilya Sutskever, Geoffrey E. Hinton·Published 3 December 2012·Computer Science·Communications of the ACM).

In 2009, Wu et al. [45] proposed a convolutional network (ECNet) for the semantic segmentation of side-scan sonar images that was fast and had few parameters. In 2016, Valdenegro et al. [46] proposed a model for object detection and recognition in forward-looking sonar images, which can also be used to detect unlabeled and untrained targets. In 2018, Hu et al. [47] proposed a new deep-learning method by combining an extreme learning machine (ELM) with a CNN in which the original acoustic signal multi-metric convolution is used to extract features in order to avoid feature defects regarding the MFCC (Mel Frequency Cepstrum Coefficient). Unlike the traditional method, the raw signal is processed in frames as an input, and the extreme learning machine classifier is chosen for the use of classifiers to achieve better results in ship-noise analysis. In 2021, Liu et al. [48] proposed a CNN-based DOA estimation algorithm for hydroacoustic arrays based on the application of convolutional neural networks in RGB three-channel image processing and proposed the use of a CNN containing real and imaginary parts of the covariance matrix of the two channels as the input signal of the CNN to estimate the direction of the hydroacoustic signal. In the same year, Mishachandar B et al. [30] found that Deep CNN has high learning capacity and adaptability when used for acoustic classification and that the Mel spectrum outperforms linear spectrograms in feature extraction; thus, an MFCC feature extraction technique was used to extract the features of acoustic spectrograms and eliminate background noise, propose new CNN networks for classification recognition, and introduce data enhancement mechanisms—we will describe the specific measures of data enhancement in detail below. In addition, dropout layers were also added to the network by the authors of [30] to reduce the overfitting of the CNN framework model.

In 2022, Guo et al. [49] applied acoustic features, Mel Frequency Cepstrum Coefficients (MFCC), and Gamma Pass Frequency Cepstrum Coefficients (GFCC) to underwater signal classification and proposed a model combining deterministic and statistical models. The geometric channel model helps to generate databases for different geometric settings, and the effectiveness of its systematic framework is verified by comparing it with continuous wavelet transform (CWT) and short-time Fourier transform (STFT) using a CNN as a classifier. The hydroacoustic signal recognition model of a CNN as a classifier is verified to have good performance and generalizability. Consequently, the discovery of a method with which to solve the inherent problems of the network due to insufficient training data has become the direction of further improvement.

Underwater targets are difficult to classify due to the influence of grazing angles, range, natural environment, and latitude and longitude. Khushi et al. (2020) [50] proposed a meta-heuristic Chimpanzee Optimization Algorithm (ChOA) based on chimpanzees' hunting behavior to train artificial neural networks. The algorithm was compared with an Ion Movement Algorithm (IMA), the Gray Wolf Optimization Algorithm (GWO), and a hybrid algorithm by comparing the convergence speed, capture localization, and performance of the different methods. By comparing the convergence speed, the possibility of capturing local minima, as well as the classification accuracy by other means, it was demonstrated that the algorithm performs better in most cases. Due to wavelength-dependent light absorption and scattering, underwater images suffer from severe color distortion and detail loss, which seriously affect underwater targets' subsequent detection and recognition. The latest methods for underwater image enhancement are based on depth models and focus on ascertaining a mapping function from the subspace of underwater images to the subspace of ground truth images, but the use of these methods often leads to different background colors of underwater images as the diversity of underwater conditions are ignored. Wu et al. (2020) [51] addressed the problem wherein assistance is needed to achieve higher accuracy after transforming a dataset into an audio spectrum. They used an improved neural network, LeNet, to fit the spectrally transformed dataset and achieved higher accuracy than the existing methods, thereby meeting the desired goal of being useful for practical applications.

### 3.2. Generative Adversarial Networks (GAN)

Goodfellow et al. [52] first introduced the GAN network, which consists of two main components, a generative model and the discriminant model, which correspond to the generator and the discriminator in the network structure. The generator is used to generate samples with an equivalent probability distribution as the actual training dataset, and the discriminator is responsible for identifying whether the input is from the actual dataset or the generator. The generator and discriminator confront each other and continuously adjust the parameters, with the ultimate goal of rendering the discriminator network incapable of judging whether the output results of the generative network are authentic. Therefore, GAN networks are often used for graph generation, data enhancement, etc. Due to the complexity of the underwater environment, GAN networks have been widely used for underwater image data enhancement.

In 2017, Juhwan Kim et al. [53] proposed an algorithm for generating genuine sonar fragments or images. The method consists of two steps: sonar image simulation and GAN-based image transformation. First, by calculating the transmission and reflection of acoustic waves, a sonar image simulator based on a ray-tracing technique is used to simulate an image containing semantic information through simple calculations. Then, the actual sonar images are transformed into simple images by adding noise or by denoising and segmentation based on GAN network principles; finally, these simple images are transformed into authentic sonar images. In 2019, Yu et al. [54] proposed a conditional generative adversarial network for underwater image recovery that used a Wasserstein GAN with a gradient penalty term as the backbone network and designed the loss function as the sum of the loss of the generative adversarial network and the perceptual loss. Unlike

standard GAN networks, the network uses a patchGAN classifier in its discriminator to learn to recognize structural losses instead of image-level losses or pixel-level losses.

In the field of hydroacoustics, GAN networks have been introduced for hydroacoustic signal recognition in order to enhance hydroacoustic data due to the practical factor that data from hydroacoustic datasets are difficult to obtain. A new framework based on generative adversarial networks (GAN), Figure 8, is proposed in [55] for addressing the problem of insufficient samples for hydroacoustic signals. The framework preprocesses audio samples into grayscale spectral images for feature capture and complexity reduction by GAN, and then evaluates the GAN-generated samples using an independent classification network external to GAN; the results show that the GAN-generated samples can significantly improve the classification and prediction accuracy of the model.

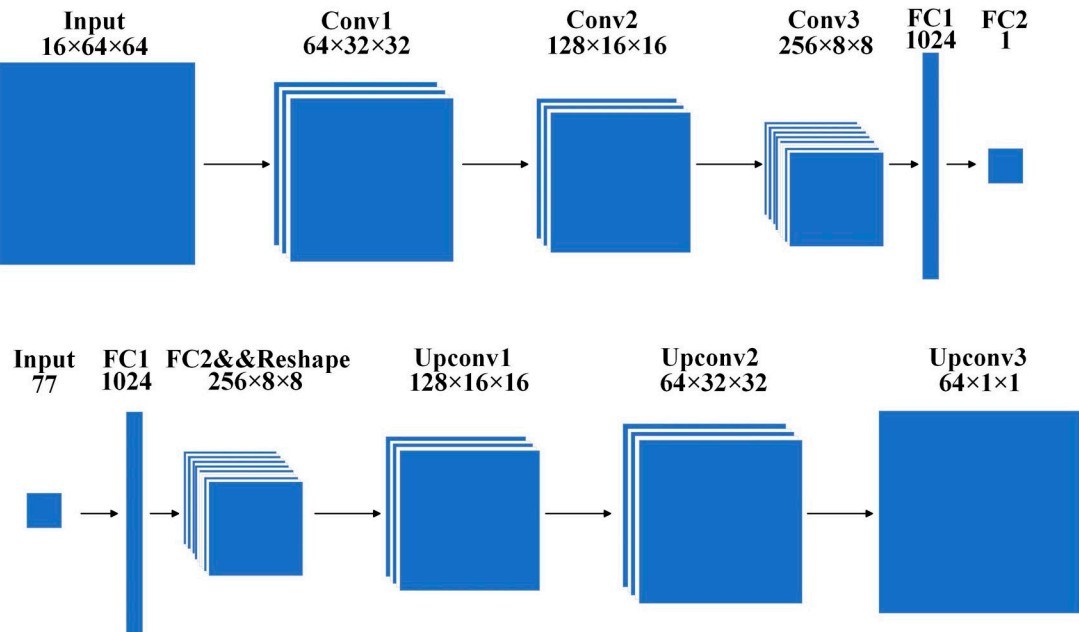

**Figure 8.** CGAN network structure proposed by Liu et al.

However, only conditional generative adversarial networks (CGAN) [56] were used in [10,55], and no denoising was performed on the original data. Using other GAN-derived models and denoising the data before transferring them to a spectral-input CNN network for recognition would probably yield better recognition accuracy.

*3.3. Recurrent Neural Networks (RNN)*

Recurrent neural networks, first proposed in 1990, are considered a generalization of recurrent neural networks, which are artificial neural networks with a tree-like hierarchical structure and network nodes that recursively respond to the input information in the order of their connections [57]. When each parent node of a recurrent neural network is connected to only one child node, its structure is equivalent to that of a fully connected recurrent neural network [58]. Since recurrent neural networks have variable topology and shared weights, they are used for machine learning tasks that contain structural relationships. They are widely used in natural language processing (NLP), speech and text recognition, and sonar recognition systems [59,60].

The authors of [25] used multidimensional features as inputs in hydroacoustic signal recognition to solve the problem of limited one-dimensional Mayer spectrum data and used a CRNN recurrent convolutional network as a classifier, which resulted in a surface-recurrent neural network with better recognition accuracy than CNNs and LSTM [61]. T. Hughes et al. [62] proposed a novel recurrent neural network (RNN) model that can properly compensate its preference for temporal continuity with acoustic features in each

frame, thereby reducing the overall speech recognition computation time by 17% while reducing the word error rate by 1%, which has implications for hydroacoustic signal recognition.

Doan et al. [63] proposed a method for underwater target recognition using a dense convolutional neural network with a network architecture designed to cleverly reuse all previous feature maps to optimize the classification rate under various impaired conditions while satisfying low computational costs; the method uses the time domain original audio signal in the time domain as an input, achieving an accuracy of 98.85% at a signal-to-noise ratio of zero, thereby outperforming conventional machine learning algorithms.

### 3.4. Transfer Learning

As mentioned above, various types of neural networks are applied in hydroacoustic signal recognition to solve the problem of difficult access to hydroacoustic data and small data sets. Another machine learning approach that can cleverly avoid the training of hydroacoustic data is the migration-learning paradigm, Figure 9, whose goal is to apply the knowledge or patterns learned on a domain or task to different versions thereof. The related domains or problems and the migration-learning paradigm are fundamental and widely accepted approaches to addressing the lack of training data in machine learning [64].

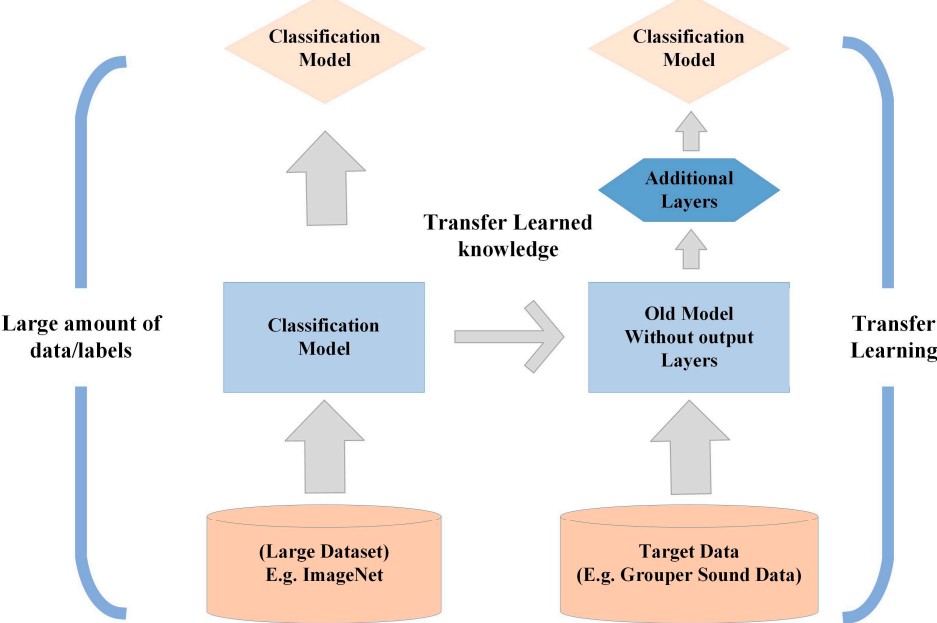

**Figure 9.** Illustration of transfer learning.

Ali K. Ibrahim et al. [65] introduced migration learning in grouper sound classification in which they processed the raw audio into spectrograms and scale maps and fed them into pre-trained deep neural network models, namely, VGG16, VGG19, Google Net, and MobileNet, and their results showed that all these pre-trained deep-learning neural networks yielded good recognition accuracy.

However, pre-training networks' pre-training weights, such as those of VGG16, are trained from visual images. Take the publicly available natural sound dataset Google AudioSet [66] as an example, which contains 632 classes of audio categories and 2,084,320 manually labeled sound clips of 10 s each. The audio ontology is identified as a hierarchical map of event categories, covering a wide range of human and animal sounds, instrument and music genre sounds, and everyday environmental sounds. In this paper, we argue that a pre-trained network trained with natural sounds can more effectively improve the accuracy of hydroacoustic signal recognition and that the migration-learning model is an efficient and valuable technical approach in case of insufficient hydroacoustic data.

## 4. Data Augmentation

In speech recognition, the data augmentation of raw audio is a typical operation whose essence is to ensure that a limited number of data produce values equivalent to more data without substantially increasing their numbers. Data augmentation has been introduced into hydroacoustic signal recognition due to the hard-to-acquire matching nature of hydroacoustic datasets and the lack of data. Next, we will introduce the traditional audio data enhancement approach and the neural network-based data enhancement approach.

### 4.1. Traditional Data Enhancement Based on Original Audio

Data augmentation is a prevalent technique in deep learning and is mainly used to enhance the training dataset such that the dataset possesses as much diversity as possible. Since hydroacoustic signals inherently face serious problems, such as a lack of data samples and insufficient sample completeness, data enhancement methods can help improve the target classification results. Data enhancement can be performed in both spectrograms and time domain signals. In the spectrogram, the enhancement problem of the acoustic signal is transformed into a more visual problem for processing, and the enhancement of the spectrogram is achieved by spatiotemporal interference and random masking techniques. This method can better cope with the distortion in the time direction and the partial loss of frequency information. Of course, the effect of spatial distortion is ignored here. In 2013, Das et al. [67] extracted spectral features and cepstrum coefficients based on spectral analysis to enhance an existing feature set. In [68], a one-and-a-half-dimensional spectrum based on a half-dimensional spectrum analyzed via the principal component analysis (PCA) method was introduced to extract ship-radiation noise. In 2016, Zhang et al. [15] extracted Mel frequency cepstrum coefficients (MFCC), first-order differential MFCC, and second-order differential MFCC features and concluded after a study that these features are the most effective with respect to acoustic target recognition regarding underwater targets' standard features. In the same year, Santos et al. [6] developed a hull classifier based on cepstrum coefficients and the Gaussian mixture model. Meng et al. [69,70] proposed zero cross-entropy features and inter-peak amplitude features to describe propeller rotation. However, their performance is significantly reduced in noisy, shallow waters. Azimi-Sadjadi [71] studied wavelet decomposition techniques. Wei [72] used wavelet features to classify underwater acoustic targets. However, it is difficult to determine the wavelet decomposition sequence using this method due to the lack of a priori knowledge. Recently, a feature extraction auditory model concerning the human auditory system has been widely adopted. Yang [73] proposed auditory features based on an evaluation of heterogeneity. However, the frequency resolution of the hydroacoustic noise they analyzed was too low to describe the details of the spectrum. Tuma and Yang et al. [74,75] fused multi-domain features of ship-radiation noise and implemented an integrated recognition system by designing a support vector machine using such extracted features. Siddagangaiah et al. [76] investigated a multi-scale, entropy-based detection method for hydroacoustic target recognition.

Since the sampling rate of acoustic signals is usually high and the time-domain signals contain limited information, manually extracted spectrograms are usually used as the input data of the network, and the recognition of hydroacoustic signals is achieved by combining manual feature extraction and deep network models. The reference VGG network is used as the base model, and some parameters in its network layers are modified to adapt to the classification task of hydroacoustic signals. The network consists of eight convolutional layers, in which each one convolves the output of the previous convolutional layer with a set of filters to capture local information in the feature map. Finally, classification is achieved by outputting the probabilities of different target classes through fully connected layers and classifiers. The deep neural network automatically extracts feature information in the acoustic spectrogram through a multilayer structure. It obtains high-level statistical features of the data through a combination of supervised linear and nonlinear data, thereby reducing manual involvement and achieving a data-driven process.

*4.2. Neural Network Data Enhancement*

The target signal in water is converted from an acoustic signal to an electrical signal or from an electrical signal to an acoustic signal by a hydroacoustic transducer. The former is the hydrophone. It is the basic device used for target detection and acquisition in water and is the basis for hydroacoustic communication, target detection, tracking, and identification. After the hydrophone acquires the signal, it is converted and saved by amplifying the analog filter.

If the acoustic signal data are collected directly via sonar to extract features and then used for classification and identification, the environmental factors affecting the signal will impact its features and the process will not achieve the desired effect. Therefore, it is necessary to pre-process the data to increase the prominence of the information in the sample data and reduce the impact on feature extraction. The main method for this is to employ pre-emphasis, a frame addition window, amplitude regularization, and the time–frequency mixing class enhancement of the hydroacoustic signal data before processing.

Firstly, pre-emphasis is adopted to reduce the attenuation of the hydroacoustic signal during high-frequency propagation, followed by frame splitting and windowing to increase the data sample size and amplitude regularization to remove DC signals so as to facilitate the subsequent feature extraction process. Due to the step involving the addition of windows in the first frame, the degree of amplitude regularization will not be normalized solely according to the maximum value of the overall signal sequence data, and the relatively small amplitude of the signal will not be changed, which plays a certain role in regularizing the hydroacoustic data in each frame. Finally, the use of time–frequency mixing class enhancement of the hydroacoustic data to expand the training model and avoid problems pertaining to the training of the model caused by an insufficient sample size leads to inadequate results, while also adding noise to the signal data to improve the robustness of the model.

If the hydrophone is compared to the human ear, the magnitude of the signal amplitude reflects the loudness information of the sound source. Since the intensity of sound in the propagation process is attenuated with distance, the "loudness" information is affected by the distance of the sound source from the hydrophone and less so by the characteristics of the target itself. However, the signal contains more information in the proportion of each frequency component of the information, indicating the relative magnitude of each frequency amplitude; thus, one can employ amplitude regulation to eliminate the influence of the source's distance to the characteristics of the impact. Amplitude regulation means that the signal amplitude is limited. Since the distance of the ship target from the hydrophone in the data set is variable, it is necessary to eliminate the influence of loudness on the characteristics.

## 5. Discussion

In deep learning, feature selection and extraction play an increasingly important role, and a good feature is sometimes even more critical than a good model. Low-level features contain more information on the original data but have undergone fewer operations and provide noisier information; high-level information contains less noise but may have lost a great deal of key information in the extraction process. The fusion of features at different scales is a new idea for feature extraction. Feature fusion methods are currently divided into two stages, i.e., fusion before the training model and fusion after the training model. Pre-training model fusion can be considered to be the parallel or serial splicing of features to increase the features' dimensions or form them into a complex variable. Then, the fused features are trained as classifiers. For example, the ParseNet model and HyperNet are designed via the serial splicing of features, and Hypercolumns use the idea of the parallel splicing of features. The process of fusion after the training of the model requires the operator to start predicting only partially fused layers before their final fusion and to then fuse the results according to the results obtained.

After the pre-processing of the data, feature extraction, and feature fusion, the selection and design of the classifier need to be performed. A convolutional neural network model is extracted and trained with respect to a two-dimensional matrix, leading to the design of a model applicable to the current hydroacoustic target recognition task. At the same time, the current hot migration-learning method is introduced into hydroacoustics, and the recognition and classification of hydroacoustic features with respect to the current dataset are performed with the help of model design ideas and structural parameters that have performed well in other fields. Based on the original hydroacoustic information and the fused features of the simulated human ear's Meer spectrum, the model is output to recognize and classify the targets in the hydroacoustic data.

## 6. Conclusions

The advent of deep-learning networks has dramatically improved the efficiency and accuracy of hydroacoustic signal recognition. With the development of network structures such as LSTM, CNN, and GAN, recognition accuracy and efficiency have been greatly improved. The latest transformer model applied in the field of NLP has shown powerful efficiency with respect to human voice recognition. Various neural networks and signal-processing approaches have been applied to hydroacoustic signal recognition, which has further improved its recognition accuracy. However, these methods still have many drawbacks that need to be improved upon. Xie et al. estimated a missing dimension in side-scan sonar using a typical residual neural network, ResNet, and a UNet network to estimate water depth contours. For the underwater targeting of the problem of a small target dataset, Jin et al. [9] applied LOFAR spectra for preprocessing to retain critical features and used generative adversarial networks (GAN) to extend the samples to improve performance classification. The experimental results showed that the generated samples had high quality and could significantly improve the classification accuracy of the neural model. The application of deep-learning models such as convolutional neural networks to hydroacoustic target recognition can significantly improve classification accuracy and constitutes a new research direction in hydroacoustic detection.

In summary, hydroacoustic target recognition technology will develop in terms of intelligence, autonomy, high accuracy, robustness, and real-time acquisition, and will play a more significant role in military and civilian fields; meanwhile, artificial intelligence and array-signal-processing technology for complex environmental motion parameter estimation, multi-target recognition tracking, and the improvement of array directivity are the promising directions and development trends of future research.

This paper summarizes the latest research progress regarding underwater acoustic target recognition and compares the accuracy of the various methods proposed in the references in an effort to ascertain the developmental directions of underwater acoustic target-recognition technology, see Figure 10.

In this paper, the following suggestions are made for hydroacoustic signal classification.

(1) The development of a signal to preprocess feature extraction is crucial, Table 1. Although short-time Fourier, Meier, Hilbert–Yellow, and other processing methods have been proposed to solve part of signal feature extraction; however, due to the shortcomings of the various algorithms, single signal-processing feature extraction can no longer improve the efficiency of the classifier. Therefore, multi-spectrum feature fusion will be one of the directions of development of hydroacoustic signal recognition.

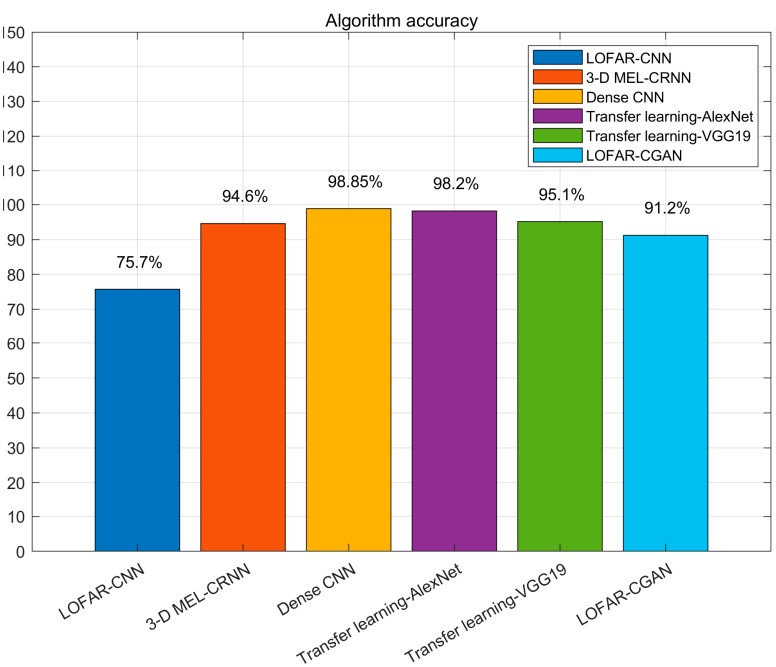

**Figure 10.** Accuracy of partial method recognition.

**Table 1.** Preprocessing methods.

| Method | Strengths/Features | Limitations |
|---|---|---|
| Short-time Fourier transform (STFT) | Obtains the signal power spectrum at different moments<br>Creates a time–frequency analysis chart of hydroacoustic signals<br>Includes multimodal fusion features<br>Highly distinguishable | Lack of time- and frequency-locating functions<br>Low time–frequency resolution of hydroacoustic signals |
| The Wavelet Transform | Features multiple resolutions<br>Reduces high-frequency interference components<br>Provides significant noise reduction effect towards hydroacoustic signals<br>Widely used in hydroacoustic field | Lack of adaptivity compared to other modal decomposition methods |
| The Hilbert Yellow Transform | Analyzes nonlinear non-smooth signals and is applicable to hydroacoustic signals<br>Suitable for mutational signals | Theoretical framework is difficult to establish<br>Endpoint effect problem exists |
| Mel-Frequency Analysis | High resolution in the low-frequency section of the hydroacoustic signal<br>Good recognition performance even when signal-to-noise ratio is reduced<br>Widely used in speech recognition | The dimensionality reduction process leads to the loss of some of the original data |
| Mel Frequency Cepstrum Coefficient (MFCC) | Combination of dynamic and static features<br>High hydroacoustic signal recognition capability<br>Widely used in speech recognition | High-frequency part is not sensitive |
| Feature fusion | Compensates for the missing features of individual spectrum features<br>More features can be extracted from a small number of training data<br>High efficiency regarding deep-learning networks | After the feature dimension reaches a certain size, the performance of the model will decrease |

(2) The improvement of classifier neural networks is necessary, Table 2. In the back-end, the efficiency of the classifier network determines the accuracy and speed of recognition. In hydroacoustic signal recognition, the back-end decision algorithms commonly used in computer vision such as random forest can be introduced. Improving the neural network's efficiency will be a critical issue.

**Table 2.** Deep-learning methods.

| Method | Strengths/Features | Limitations |
|---|---|---|
| Convolutional Neural Network (CNN) | Convolution layer enables feature extraction for easy feature extraction of spectrograms<br>Handles high-dimensional data<br>Highly versatile | A great deal of valuable information will be lost<br>Large number of labeled training data are required<br>Contradictory to the lack of hydroacoustic data |
| Generative Adversarial Network (GAN) | High unsupervised learning ability<br>Suitable for small data sets | Generate single data<br>Low network ubiquitousness |
| Recurrent Neural Network (RNN) | Widely used in text and speech analysis<br>The mathematical basis can be considered as Markov chains with memory capacity | Unable to support long sequences<br>Cannot distinguish between ambient noise and ship noise |
| Transfer learning | High learning capability<br>No reliance on large data sets | Reliant on pre-trained networks<br>Less hydroacoustic data leads to inadequate pre-trained network |
| Temporal Convolutional Network (TCN) | Training is applied directly through the original audio<br>Has applications in speech recognition | Unable to handle noise in an aquatic environment<br>Low accuracy in recognition of hydroacoustic targets |

(3) Hydroacoustic data enhancement is essential, Table 3. Due to the complexity of the marine environment, marine environmental noise varies significantly in different sea conditions, different sea areas, and at different times. Improving classification models' generalization via data enhancement is a problem to be solved.

**Table 3.** Data augmentation methods.

| Method | Strengths/Features | Limitations |
|---|---|---|
| Traditional data augmentation methods (audio editing and synthesis, etc.) | Simple operation | Highly dependent on original audio data |
| Neural Network data augmentation | Ability to handle unrelated features<br>Suitable for processing samples with missing attributes<br>Compensates for lack of hydroacoustic data | Ignores correlation between data |

(4) The small sample problem must be solved. Notably, the application of deep-learning models such as convolutional neural networks to hydroacoustic target recognition can significantly improve classification accuracy and constitutes a new research direction in the field of hydroacoustic detection, which will lead to the improvement of performance with respect to faint signal detection as well as underwater target identification and localization. However, the computational complexity of these algorithms needs further attention. On the other hand, hydroacoustic targets usually combat divers, underwater crewless vehicles, submarines, etc., which have a certain degree of concealment and confidentiality. Thus, it is more difficult to obtain their target database. While data-driven deep-learning based on a large number of data samples is required for training, the small sample problem also needs to be addressed.

**Author Contributions:** X.L. conceived of the study, designed the study, and wrote the manuscript. Z.L. provided theoretical guidance and revised the manuscript. R.D. assisted in the editing of manuscripts. All authors have read and agreed to the published version of the manuscript.

**Funding:** This research was funded by Shandong Province "Double-Hundred" Talent Plan (WST2020002) and the Open project of the State Key Laboratory of Sound Field Acoustic Information (No. SKLA202203).

**Data Availability Statement:** Not applicable.

**Conflicts of Interest:** The authors declare no conflict of interest.

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
