# Peer review of "Deep Learning-Based Classification of Raw Hydroacoustic Signal: A Review"

_jmse, doi:10.3390/jmse11010003_

Round 1

Reviewer 1 Report

This paper provides a comprehensive overview of the deep learning methods for analyzing hydroacoustic data. It is a well-written manuscript on an emerging topic, covering all the necessary details on the current literature. It was extremely pleasant to read this manuscript that presents a complex material in a organized and easy-to-understand way. I recommend the publication in the current form and would like to see it in the current literature.

Reviewer 2 Report

Overall, this review manuscript examines a very relevant topic. However, I do have the following comments that need to be addressed before being accepted for publication.

1.      It is important to mention the main finding of the paper in the abstract. This is the section that gains the attention of the audience and needs to highlight the most important findings from the study.

2.      I would recommend referencing the following manuscript in the introduction: “Chronic leak detection for single and multiphase flow: A critical review on onshore and offshore subsea and arctic conditions” by Behari et. al, as it is a review paper that examines the problem from a broader point of view and may be of interest to members of the audience.

Reviewer 3 Report

This paper presents a review of Deep Learning-based Classification of Raw Hydroacoustic Signal. The review is not comprehensive and lacks detailed critical analysis and discussion. The authors have outlined several topics, however they did not carry out critical assessment covering issues, challenges, concerns, key findings, contributions and future suggestions. Thus it is highly suggested to summarize all the key outcomes and analysis into 3-4 tables. Also, it is recommended to classify deep learning approaches and then discuss their contributions. In conclusion, I advise the authors to rearrange the paper and focus on critical discussion. 

Reviewer 4 Report

The proposal is very general, critical and/or minimum data are not presented for each section, the contribution that the authors wish to make is not clear, given that as a review it is very poor and as a proposed solution there are no data to validate it. In case a major revision is allowed, an improvement in the proposal is required, indicating the contribution with theoretical analysis and data to be evaluated.

Round 2

Reviewer 3 Report

The authors have revised the paper comprehensively. The paper can be accepted. 

Reviewer 4 Report

The new proposal still lacks minimal theoretical analysis and simulation or experimental data to understand the contribution. I consider that for its possible publication, the article proposal should be improved, where the significance of the contribution should be clearly indicated.

Round 3

Reviewer 4 Report

The authors have considered the recommendations, I consider that it presents a sufficient contribution for publication.